# Molecular Pathways Related to Sulforaphane as Adjuvant Treatment: A Nanomedicine Perspective in Breast Cancer

**DOI:** 10.3390/medicina58101377

**Published:** 2022-10-01

**Authors:** María Zenaida Saavedra-Leos, Euclides Jordan-Alejandre, Jonathan Puente-Rivera, Macrina Beatriz Silva-Cázares

**Affiliations:** 1Academic Coordination of the Altiplano Region, University Autonomous of San Luis Potosi, San Luis Potosi 78700, Mexico; 2Genomics Sciences Prograde, Autonomous University of Mexico City, Mexico City 03100, Mexico; 3Division of Health, Biological and Environmental Sciences, Open and Distance University of Mexico, Mexico City 03330, Mexico

**Keywords:** breast cancer, sulforaphane, nanomolecule

## Abstract

Because cancer is a multifactorial disease, it is difficult to identify the specific agents responsible for the disease’s progression and development, but lifestyle and diet have been shown to play a significant role. Diverse natural compounds are demonstrating efficacy in the development of novel cancer therapies, including sulforaphane (1-isothiocyanate-4-(methylsulfinyl)butane), a compound found in broccoli and other cruciferous vegetables that promotes key biological processes such as apoptosis, cell cycle arrest, autophagy, and suppression of key signalling pathways such as the PI3K/AKT/mTOR pathway in breast cancer cells. However, one of the primary challenges with sulforaphane treatment is its low solubility in water and oral bioavailability. As a consequence, several investigations were conducted using this component complexed in nanoparticles, which resulted in superior outcomes when combined with chemotherapy drugs. In this study, we discuss the properties and benefits of sulforaphane in cancer therapy, as well as its ability to form complexes with nanomolecules and chemotherapeutic agents that synergize the antitumour response in breast cancer cells.

## 1. Introduction

Breast cancer is the most frequent malignancy in women and the second most common cancer worldwide, characterized by the aberrant proliferation of cells and deregulation of cell cycle control [1,2,3]. Breast cancer is thought to be caused by a mix of genetic and lifestyle variables, but the extent to which an overall healthy lifestyle can reduce the influence of many genetic variations on the risk of invasive breast cancer is unclear. Advances in epidemiological and clinical breast cancer research have allowed researchers to identify a number of risk factors linked to women’s reproductive history and lifestyle such as food, body weight, and physical exercise. Obesity and being overweight in post-menopausal women have been associated with an increased risk of breast cancer. At the same time, protective factors such as physical exercise, breastfeeding, and a well-balanced diet can significantly reduce the risk of breast cancer [4]. In this perspective, it is clear that food plays a dual function as a protective or risk factor. In fact, nutritional oxidative stress can be caused by an imbalance between antioxidant response and pro-oxidant load as a result of insufficient or excessive food availability [5,6,7]. The analysis results of the various nutrients or bioactive chemicals found in vegetables for cancer prevention is becoming increasingly important, since these studies have revealed a favourable influence on tumour growth and development by inducing apoptosis and decreasing cell proliferation [8]. In general, certain bioactive and other compounds can suppress or promote the expression of certain key genes in breast cancer by modulating their transcription, blocking or activating signalling pathways involved in proliferation, differentiation, and apoptosis, while counteracting the toxic effects of chemotherapy drugs. For example, Bober et al. pointed to an assumed role of doxorubicin (DOX) in dysregulation of actin cytoskeleton and cell death [1,9,10,11,12,13].

## 2. Effects of Glucosinolates Derived from Cruciferous Vegetables

In 2013, an epidemiological study with a cohort of 69,120 participants described one of the first approaches to the relationship between dietary patterns and cancer development. The dietary patterns considered different food ingestion, and the results suggested a protective effect of a vegetarian-like diet showing a protection for overall cancer incidence, and interestingly, the lacto-ovo-vegetarian regiment appears to be a gastro-intestinal prevention factor [14]. Various small phytochemicals, especially dietary phytochemicals such as sulforaphane, mahanine, resveratrol, linolenic acid, diallyl sulfide, benzyl/phenethyl isothiocyanate, etc., have been considered in line with the recent advancement in the acceptance of these types of potential dietary phytochemicals as chemo-preventive agents against the development of breast cancer [15]. These relations between vegetarian-like diets could explain the major ingestion of vegetal antioxidants molecules; Razis et al. described the role of cruciferous vegetables as cancer protection factors specifically the glycosylates—they described the relation between these molecules and the reduction of different cancer types like colorectum, lung, prostate, and breast [16]. The prevention factor related to vegetarian diets is constituted in part by the consumption of cruciferous vegetables such as Brussel sprouts, cabbage, cauliflower, collard greens, kale, kohlrabi, mustard, rutabaga, turnips, bok choy, and Chinese cabbage; results of different investigations suggested that the anticancer effect is related to the high concentrations of glucosinolates [17]. Broccoli is a vegetable related to the Brassica oleracea group (var. Italica) [18]. These vegetables have multiples benefits for the healthy. In different studies, the authors related a broccoli-enriched diet as a preventive factor for diseases such as cancer and neurodegenerative diseases, while diabetes studies revealed the prevention of complications and inclusive randomized studies revealed the reduction of autism symptoms [1,19,20,21,22]. Broccoli is an economical vegetable whose growth occurs at 18 °C to 25 °C [23]. Studies suggest that broccoli can prevent and improve the treatment of diseases, especially diseases related to the oxidative and inflammatory process because it produces antioxidative molecules.

The glucosinolates are a class of sulphur containing glycosides, common glucosilates are glucoraphanin, glucoerucin, glucoraphasatin, and gluconasturtiin among others—these glucosinolates are found in broccoli, recked salad, radish and watercress respectively [16]. Such molecules play an important role in cancer prevention, although they could be activated by hydrolysis—myrosinases are enzymes capable of ‘activating’ glucosinolates. Furthermore, these enzymes activate molecules when vegetable tissue is cut or chewed, generating a substance known as isothiocyanate [24]. Sulforaphane (1-isothiocyanato-4-(methylsulfinyl)butane) is an isothiocyanate generated by hydrolysis of isothiocyanates by myrosinases enzymes; the process occurs when mastication disintegrates the myrosinase compartments. The interaction between the enzyme and the glucosinolates generates hydrolysis of the thioglucosidic bond, digestion generates two different products, (I) glucose and (II) unstable aglycone while the next step generates different products (Table 1) depending on the physicochemical conditions generated in the first step [25].

There is a stable correlation of isothiocyanate formation with a pH range of 6–7, despite the fact that glucosinolates possess a β-hydroxy-isothiocyanate that being unstable spontaneously cyclises to oxazolidine-2-thiones (e.g., goitrin) and indole isothiocyanates after lysis generates the corresponding alcohol as indole-3-carbinol. In contrast to a pH between 4 and 7 the presence of the nitrile generates principally ascorbigen and thiocyanate, nevertheless the mechanism of ‘nitrile-forming factor’ is not clear [35].

## 3. Apoptosis, Cell Cycle, Autophagy, and Suppression of the PI3K-AKT-mTOR Pathway Are the Key Biological Processes Influenced by Sulforaphane

### 3.1. Sulforaphane Promotes Mitochondria-Mediated Apoptosis

The main method of treatment related to cancer involves increasing the susceptibility of apoptosis in cancer cells. Sulforaphane treatment studies provide a consistent mechanism of action that is related to a reduced mitochondrial membrane potential that promotes apoptosis in cancer cells and tumour xenografts. Cancer research shows that sulforaphane treatment increases caspase-9 and poly-ADP ribose polymerase (PARP-1) activity, as well as cyclooxygenase IV (COX IV) activity, sensitizing cancer cells to mitochondria-mediated apoptosis [36]. Interestingly, the apoptotic effect of sulforaphane is also mediated by the receptor type 1 inositol 1,4,5-trisphosphate (IP3R1) known to promote mitochondrial Ca^2+^ overload leading to cell apoptosis 24 h after treatment. Additionally, sustained treatment for 7 days led to a significant reduction in tumour size observed in mice tumours [37]. Remarkably, the combination of Lactobacillus and sulforaphane could enhance apoptotic effects and reduce the cell growth of colon cancer cells HCT116 and SW480 [38]. The study shows that the up-regulation of pro-apoptotic factors such as *TNFR1*, *cIAP-1*, *cIAP-2, Bax,* and mitochondrial membrane potential decreased after co-treatment, suggesting apoptosis activation [38]. On the other hand, Myzak et al. showed that after 48 h of sulforaphane treatment, an increase in acetylation in the promoter region of p21 and Bax is produced by HDAC, triggered cell cycle arrest and activation of multicaspase activity [39]. Caspase activity was also detected in H727 and H720 cells treated with sulforaphane, showing in treated cells a cleaved caspase-3 in 70%, caspase-7 in 89%, and cleaved PARP in 113% which triggered the induction of apoptosis in 53% of treated cells after 2 weeks of sulforaphane treatment [40]. Other experimental investigations demonstrated that sulforaphane therapy promotes the expression of a lysosome-associated membrane protein 2 (*LAMP2*), which decreases sulforaphane’s capacity to trigger apoptotic cell death, at least in prostate cancer. *LAMP2* knockdown enhanced apoptosis in PC-3 and 22Rv1 cells treated with 20 μm/L sulforaphane. Finally, another mechanism of apoptosis related to sulforaphane is the activation of p34cdc2 kinase by dephosphorylation—the activation induced apoptosis in 25% and 35% of treated cells after 24 and 48 h. In addition the authors suggested that the induction of the proteasome by ubiquitination is essential to induce the mechanism of apoptosis in treatment with sulforaphane [41].

### 3.2. Effects of Sulforaphane Treatment on Cell Cycle

Understanding oncogenesis and apoptosis reveals the critical function of cell cycle control in malignant transformation. When normal cells acquire DNA damage or mutations, checkpoints are activated to stop cell cycle progression and promote DNA repair or induce cell apoptosis. For example, Sundaram et al. showed apoptotic changes such as nuclear condensation, fragmentation, and formation of apoptotic bodies probably due to the induction of *NOS2* and *NOS3* expression, triggering increased nitric oxide. Furthermore, the study reveals the upregulation of *HSP-90AB1*, *PRKAR1B*, *ALOX12*, *PRG3,* and *NCF2*. On the other hand, sulforaphane treatment led to downregulation of genes such as *CCNA1*, *SOD3,* and *GPX4* among other genes related to maintenance of redox balance [42]. However, cancer cells often lack these regulatory systems, resulting in uncontrolled cell cycle progression and proliferation. Interestingly, multiple reports have shown that sulforaphane has an influence on the cell cycle in cancer cells. For example, Myzak, and Dashwood, and Hao et al. described cell cycle arrest of at least 18.5% (in control contrast) in phase G0/G1, triggered by sulforaphane treatment for 24 h. Furthermore, the percentage of cells inducing apoptosis was 43% higher compared to the control and the maximum rate observed was 106% higher in the treated group compared to the control [43]. In addition, sulforaphane treatment decreased migration and inhibited the metastasis process [44]. Another similar study showed that treating HCT116 cells with 20 µm of sulforaphane promote cell cycle arrest and increased apoptosis. These effects were associated with blocking of cyclin B1, Cdc25B, and Cdc25C 24 h after treatment. Although the results do not show an impact of sulforaphane on Cdk1, it was observed that increased Cdk1 on Tyr15 residue phosphorylation, inhibiting in this way the activation of the Cdk1/Cyclin B1 complex; the treatment induces the phosphorylation of the Ser216 residue Cdc25C and the subsequent bind of 14-3-3β favours translocation of Cdc25C to the cytoplasm. Furthermore, the results suggested the activation of Chk2 by phosphorylation of Thr-68 1 h post-treatment, notwithstanding the effect on Chk1 was not significant—this effect was maintained even 24 h after withdrawal of the sulforaphane treatment [45].

Other research has shown sulforaphane might decrease the expression of cell cycle proteins such as Cyclin D1, Cyclin A, and C-Myc 4 h after treatment with 50 μM of sulforaphane in HT-29 cells. In contrast, higher expression of cell cycle inhibitory proteins such as p21 was reported. Interestingly, the MAPK pathway, ERK, JNK, and p38 activation were linked with sulforaphane therapy. However, the molecular mechanism that permits inhibition of cell growth by these pathways and proteins is not well known since they are contradictorily connected with cell growth [46]. P21 expression seems to be one of the factors most related to sulforaphane therapy. Transcriptomic research has shown that 50 μM of sulforaphane enhances the restoration of p21 levels in Caco-2 cells. Transcriptomic analysis also found 169 genes with differential expression, with 106 genes increasing and 63 decreasing, suggesting that sulforaphane is a potential molecule with numerous molecular targets which might be studied for a better understanding of cancer therapy with sulforaphane [47]. Parnaud et al. also showed p21 over-expression and G2/M arrest after sulforaphane treatment in HT29 cells. Additionally, the treatment promoted the phosphorylation of retinoblastoma protein tumour suppressor after 24 h having the maximal expression after 48 h of treatment favouring apoptosis induction [48].

### 3.3. Sulforaphane Treatment Promotes Autophagy

Cell autophagy is a tightly controlled catabolic process by which cells recycle their internal components by sending them to lysosomes. Autophagy has been shown in several studies to perform a broad range of physiological and pathological functions in cells. Autophagy acts as a tumour suppressor in cancer, enabling cells to eliminate damaged cellular contents, consequently reducing ROS and DNA damage [49]. In terms of autophagy, a rising number of studies demonstrate that sulforaphane enhances its activation in cancer cells. In squamous cell carcinoma, an autophagosomal protein known as LC3B-II was shown to be increased until it peaked at 24 h after sulforaphane therapy, promoting the development of autophagosomes and auto-lysosomes. Surprisingly, the action of autophagy could be connected to the decrease in tumour weight of in vivo models after therapy [50]. Reports in prostate cancer show similar results, in relation to the overexpression of LC3-BII in treated cells with 20 μM/L of sulforaphane. In addition, microarray analysis showed 25 genes related to the autophagy regulation process in sulforaphane treatment after 6 to 9 h including only 13 genes identified over the entire period, namely *HSP90AA1*, *MAP1LC3B*, *MAP1LC3A*, *EIF2AK3*, *HSPA8*, and *UVRAG* which are upregulated and *ATG4C*, *FAS*, *PTEN*, *ATG10*, *PRKAA1*, *TP53*, and *NF-κB 1* as downregulated genes [41].

### 3.4. Sulforaphane Treatment Has Negative Effects on the PI3K-AKT-mTOR Oncogenic Signalling Pathway Preventing Carcinogenesis and Angiogenesis Process

The PI3K/AKT/mTOR signalling pathway is constitutively active in several cancer processes and plays an important role in carcinogenesis and development. Second, the PI3K/AKT/mTOR pathway regulates cancer cell survival, proliferation, migration, and therapy response [51]. A number of studies have shown that sulforaphane suppresses the mTOR signalling pathway and induces autophagy [40]. Particularly, phosphorylation of AKT, mTOR, ribosomal protein S6 kinase, and eukaryotic translation initiation factor 4E binding protein 1 were inhibited by sulforaphane and linked to a substantial decrease in tumour growth in mice after 24 days of therapy [36]. Sulforaphane co-treatment with chemotherapeutic drugs such as acetazolamide inhibits the PI3K/AKT/mTOR signalling pathway, which interacts synergistically to induce apoptosis. In both in vitro and in vivo xenograft tissues, this combination decreased tumour cell survival relative to each drug alone. Both H727 and H720 cell treatments were associated with the induction of apoptosis, elevation of the p21 cell cycle inhibitor, and downregulation of the PI3K/Akt/mTOR pathway, indicating that the PI3K/Akt/mTOR pathway is a primary target of the acetazolamide + sulforaphane combination treatment [40]. On the other hand, angiogenesis is considered to play a key role in the development and progression of cancer, generating vascularized solid tumours with a high micro-vessel density. Recent research reveals that tumour angiogenesis includes signalling cascades between tumour cells and the stromal microenvironment, resulting in the creation of aberrant vasculature, which contributes to tumour growth and metastasis. Liu et al. analysed in liver cancer the effect of 20 μM of sulforaphane, demonstrating an essential role in inhibiting STAT3/HIF-1/VEGF signalling and promoting anti-angiogenesis effects [52]. Another mechanism described for sulforaphane anti-angiogenic activity is described by Davis et al. Their data suggest FOXO activation triggers cell migration and capillary tube formation inhibition [53]. Further, angiogenesis is involved in the exacerbation of hypoxia by activating the expression of hypoxia-inducible factor 1 (HIF-1), which is connected with cancer and angiogenesis. Many genes, including vascular endothelial growth factor (VEGF), inducible nitric oxide synthase, and lactate dehydrogenase A, are regulated by it. Surprisingly, sulforaphane was reported to decrease hypoxia-induced HIF-1 expression by activating the JNK and ERK signalling pathways. The inhibition of HIF-1 by sulforaphane resulted in a decrease in VEGF expression. These findings indicate that sulforaphane is a potent chemo preventive agent against angiogenesis in vitro in cancer cells, suggesting that the HIF-1 target throws light on the processes behind sulforaphane’s suppression of tumour cell growth [54]. Different molecular pathways are summarized in Figure 1.

## 4. Tumour-Suppressive Effects of Sulforaphane in Breast Cancer

Breast cancer is the most prevalent cancer in women, with an incidence rate that has increased by 0.5% per year. In the case of breast cancer, the mechanism of action of sulforaphane is multi-targeted, as we show in Table 2, from apoptosis to autophagy, as demonstrated in other cancer types.

Sulforaphane’s anticancer activity seems to be dose-dependent in breast cancer. For example, sulforaphane of 40 µM promotes the induction of early/late apoptotic and necrotic cells in MDA-MB 231. Down-regulation of genes involved in the epithelial-mesenchymal transition, such as *ZEB1*, *claudin*, and *fibronectin*, was seen at doses of 20, 30, and 40 μM at 72 h after sulforaphane therapy [40]. Castro et al. observed a similar impact on triple negative breast cancer cells MDA-MB-231 with a 45% reduction in cell growth at 15 μM of sulforaphane. Additionally, following five weeks of treatment with 50 mg/kg sulforaphane, female BalbC/nude mice demonstrated a 29% decrease in tumour volume. The research also indicated that after 36 days of therapy, the tumours’ transcriptomes showed downregulation of stem-cell-related genes such as aldehyde dehydrogenase 1A1 *(ALDH1A1)*, *NANOG*, *GDF3*, and the embryonic pluripotency-maintaining transcription factor Forkhead box D3 (*FOXD3*) [55]. Consistently with previous results on the effect of sulforaphane under cell cycle, Royston et al. demonstrated the same effect in two different breast cancer cells line, MC7-cells and MDA-MB-231 cells; the study reported the downregulation of Cyclin 1 (*CCND1*) and *CDK4* expression promoting cell cycle arrest in G1/S phase progress [56]. The impact of sulforaphane differs depending on the cell line investigated, with the IC50 for three distinct lines varying among them, 4.05 M, 19.35 M, and 16.64 M for MCF-7, MDA-MB-231, and SK-BR-3 cells, respectively. However, comparable effects such as cell cycle arrest and the generation of genotoxic species such as reactive oxygen and nitrogen species were reported. Furthermore, sulforaphane treatment caused both double-strand and single-strand DNA breaks only in MDA-MB-231 cell line [57]. Finally, sulforaphane treatment is capable of stopping the “vicious cycle” of osteoclast differentiation that often occurs in breast cancer. Results led to the fact that sulforaphane can negatively regulate the transcription factor RUNX2, which also led to up-regulation of the *NF-κB 1* gene, suggesting that sulforaphane indirectly influences the regulation of the NF-κB pathway; These results were replicated by the same research team on in vivo models observing a 30–52% decrease in plasma levels of certain proteins such as CTSK, RANKL, and IL8 [58].

## 5. Sulforaphane as Potential Chemotherapy Adjuvants in Breast Cancer

Currently, the combination of chemotherapeutic medications with various natural molecular components has shown multiple advantages such as improved effectiveness and reduced adverse effects. In the case of sulforaphane, considerable benefits have been shown when combined with several chemotherapeutic drugs in basic scientific study (Table 2). For example, in the MDA-MB-231 breast cancer cell line, combining 5-fluorouracil and sulforaphane increases autophagy by increasing LC3-II protein, resulting in decreased cell growth and an increase in apoptosis after 72 h [68]. Another chemotherapeutic that has been studied in combination with sulforaphane is DOX. HER2 breast tumours and metastatic disease are targets for the chemotherapy DOX. However, cardiotoxicity is one of the most common adverse effects. During DOX therapy, sulforaphane at a dosage of 4 mg/kg as an adjuvant helps reduce cardiotoxicity by enhancing mitochondrial activity. In addition, sulforaphane + DOX showed significantly greater tumour regression than sulforaphane or DOX alone [69]. Additional investigation, shows that sulforaphane + DOX reduces mRNA expression of COX-1 and COX-2, recognized inflammatory mediators that promote tumour cell motility and invasion. The co-treatment impact of DOX and sulforaphane on tumour development in the BALB/c mice tumour model is equivalent to the in vitro model [70]. In general, neoadjuvant treatment for TNBC patients comprises PTX or DTX chemotherapeutics. However, PTX or DTX therapy causes IL-6 release and the proliferation of cancer stem cells (CSC) in breast cancer cell lines. Burnet et al. demonstrated that sulforaphane co-treatment (2.5 mM–15 mM range) reduces the number of CSCs by suppressing NF-κB and reversing aldehyde dehydrogenase positive (ALDH+) enrichment caused by DTX [71]. PTX-sulforaphane combination has also been shown to be effective in MCF7 cells with luminal subtypes and MDBA-MB-231 cell lines. The findings indicate that treatment with 5 M sulforaphane and 10 nM PTX for 24 h activates cell cycle progression in G1, reducing viability in breast cancer cells. Furthermore, the results indicate that apoptosis was activated by an increase in caspase 8, 9, and 3 cleavages and cytochrome C releases [57,72]. The study of Zhang et al. showed the suppression of metastasis in triple-negative cancer cell lines, independently of their chemical conformation. Their results demonstrated the activation of different pathways by the induction of toxic response (Figure 1). Here they demonstrated that the sulforaphane is capable of inhibiting the induction of migration by TGF-β1, and furthermore they demonstrated a change expression profile and a phosphorylation impact of sulforaphane on 15 proteins and 128 proteins respectively. Interestingly the new actin fibre formation was suppressed by the treatment and the compound showed similar results in other proteins related to the migration process—probably the compound could act by blocking the RAF phosphorylation action about MERK and ERK proteins, arresting the RAF/MEK/ERK signalling pathway [59].

## 6. Anti-Tumour Effects of Sulforaphane Nanoparticles: Promising Chemotherapeutic Adjuvants in Breast Cancer

Nanotechnology has been actively researched and used in cancer therapy as nanoparticles potentially play an important role as a drug delivery method. Nanoparticle-based drug delivery provides distinct benefits over traditional drug delivery, including greater stability and biocompatibility, increased permeability and retention effect, and precision targeting. The implementation of nanoparticles for sulforaphane delivery systems might overcome its primary limitations, which are its poor solubility in water and oral bioavailability. Not surprisingly, new perspectives of sulforaphane administration through nano-composite-based therapies have promising results. For instance, the use of nanoparticles of selenium, tellurium, gold-coated Fe_3_O_4_, PEGylated Fe_3_O_4_, monomethoxypol, and based on Fe^2+^ and Fe^3+^ for the administration of sulforaphane in breast cancer, has recently been documented [66,67,73,74,75]. According to the findings of these studies, the anti-tumour effects of nanoparticle treatment are capable of decreasing the viability of tumour cells such as MDA-MB-231, MCF-7, and SKBR-3. Interestingly, most of these studies support the activation of apoptosis in tumour cells by promoting the upregulation of pro-apoptotic *BAX* and *Bak* and the downregulation of anti-apoptotic *bcl-2* and *bcl-xL.* Furthermore, sulforaphane-coated selenium and tellurium nanoparticles appear to specifically target tumour cells, whereas normal cells, MCF-10A, have a better tolerance to treatment, as observed in the MTT assay [61,66,67]. Sulforaphane nanoparticle therapy in breast cancer has consistently anti-tumour effects and therefore its adjuvant administration in combination with other chemotherapeutic drugs through nanoparticles offers great potential.

### Potential Effects of Sulforaphane Nanoparticles as Adjuvant

According to the findings described in breast cancer lines like MDA-MB-231 and MCF7, the combination of sulforaphane + DOX was entrapped in liposomes nanoparticles at 9.375 mg/g of DOX, while for sulforaphane, it was 487.5 mg/g. Cell viability was statistically lower after the application of the DOX/sulforaphane nanoparticle combination. In addition, cell viability is reduced by 50% when sulforaphane and DOX are combined and seem to be effective in terms of their ability to inhibit tumour cell growth (Figure 2). Additionally, sulforaphane + DOX nanoparticles induce described anti-tumour mechanisms such as ROS generation, mitochondrial damage, and autophagy [61]. Sulforaphane micro-emulsion nanoparticle is another technique employed in combination with PTX. When utilized at high doses (8.7 µM of sulforaphane and 500 nM of PTX), the sulforaphane emulsion promotes the solubilization of PTX and improves its activity; moreover, a substantial growth suppression was found in PTX-sulforaphane micro-emulsion treated MDA-MB-231 and MCF-7 cell lines (Figure 2). PTX is a highly potent chemotherapeutic drug against breast cancer cells, however, it contains excipients, which cause several of the adverse effects. Sulforaphane, on the other hand, decreases the content of excipients utilized in commercial PTX preparations [62]. On the other hand, it has been reported that the methoxy poly (ethylene glycol)-poly (lactide-co-glycolide) nanoparticle loaded with sulforaphane at 48.97 µM increases the chemosensitivity of cisplatin at 72.59 µM by enhancing platinum binding to DNA. Cisplatin induced moderate apoptosis in MCF-7 cells, with a 15.2% apoptotic rate. In contrast, the combination of sulforaphane + cisplatin nanoparticles increased the apoptotic rate to 39.5%. Furthermore, the sulforaphane + cisplatin nanoparticle increased the expression of p-H2AX, p53, cleaved PARP, and promoted the decrease of Bcl-2 expression, indicating that apoptosis is the primary mechanism of action (Figure 2). In vivo models showed a 74.1% reduction in tumour growth treated with sulforaphane + cisplatin nanoparticle [63]. Previous research has shown that poly (D, L-lactideco-glycolide)/hyaluronic acid nanoparticles loaded with 5 nM DTX and 350 µg sulforaphane have a novel effect on breast cancer stem cells. After 72 h of treatment, sulforaphane + DTX nanoparticle decreased the viability of MCF-7 MS cells with CD44 + CD24-epithelial-specific antigen phenotype. Both b-catenin and cyclin D1 levels were reduced, leading to cell growth inhibition (Figure 2).

A decrease in tumour growth from day 24 to day 36 was observed in the treated group, and in conjunction with these results, the nanoparticle showed anticancer activity in stem cells, with approximately 140 mammospheres per 2000 cells formed per 2000 cells compared to 210 mammospheres formed per 2000 cells in the untreated groups [64]. The research by Keshandehghan et al. describes that carbon nanoparticles loaded with sulforaphane, and metformin reduced markers associated with breast cancer stem cells such as WNT-1, -catenin, and CD44. Surprisingly, sulforaphane + metformin-induced apoptosis enhanced BAX levels. This might explain why MCF-7 and BT-474 cells are sensitive to sulforaphane + metformin nanoparticles at 48 and 42 nM, respectively, resulting in cell viability decrease. Apoptosis increased in 46% of the treated cells after 8 h of treatment [65]. Other investigation performed by Gu et al. described the potential anticarcinogenic sulforaphane effect in in vivo models. They prepared a nanocarrier with a higher capacity (33.64%) to load sulforaphane, their nanocarrier consisted of a mineralized hyaluronic acid-SS-tetradecyl, after 8 h in in vitro studies the system was capable of releasing most of the sulforaphane (>60%) at acid conditions. In vivo analyses showed the rapid accumulation and sulforaphane release in situ after 6 h, the localization increased after 24 h post injection. Furthermore, the sulforaphane concentration was higher at the excised tumour in comparison with other organs, demonstrating CD44 targeting specificity and a low toxicity system [76].

## 7. Conclusions

Sulforaphane is a relatively accessible and easy-to-acquire molecule, compared to other organic components that can be used to treat cancer. Since its discovery, it has been identified as a potent phytopharmaceutical capable of altering the resistance of carcinogenic cells. Although more studies are still needed to fully understand the molecular processes that this molecule triggers, the experimental evidence generated and compiled in this review demonstrates that in addition to being a compound that is consumed indirectly in the diet, it is a component with a high specificity for cancer cells. The experimental evidence generated and compiled in this review demonstrates that in addition to being a compound that is consumed indirectly in the diet, it is a component that presents a high specificity for cancer cells, thus proving to be a potential candidate for the treatment of different cancer types. Additionally, its efficiency when complexed with various types of nanoparticles improves the prospects for its use in the development of targeted and specialized therapies for each patient. Apart from this evidence, it has also been observed that the use of sulforaphane in conjunction with strategies currently implemented in the treatment of cancer enhances their therapeutic effects, also improving the tolerability of the side effects in these therapies. In this regard the clinical studies that exist regarding the use of sulforaphane in the treatment of breast cancer, have not yet obtained a nanoparticulate as a candidate for phase 3 studies and subsequent therapy. However, overall, the experimental studies demonstrate a high safety and bioavailability of the nanocomplexes studied by different authors, each with different perspectives and different chemical and physical characteristics that allow new perspectives for the use of these nanoparticles to be generated. These can be focused not only on the controlled release of sulforaphane but also in approaches that involve treatment in conjunction with physical methods. The research of new molecular targets and signalling pathways of sulforaphane will allow the development of its clinical applications for the prevention and treatment of breast cancer. In the evidence described in this review, we have emphasized the antioncogenetic sulforaphane role and its possible use in cancer treatment as adjuvant or complexed with different nanomaterials.

## Figures and Tables

**Figure 1 medicina-58-01377-f001:**
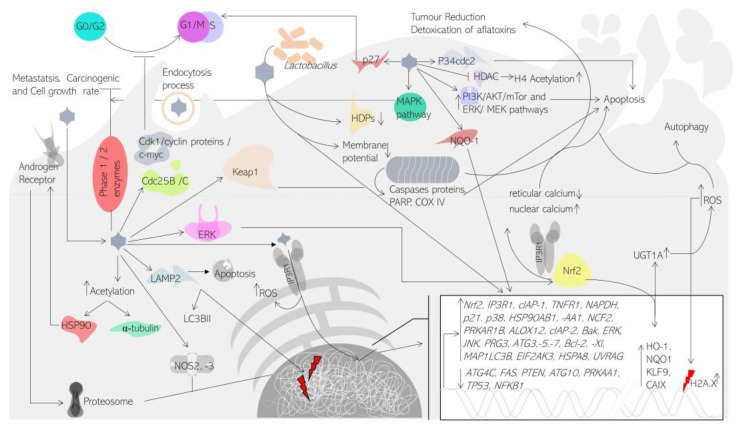
Molecular pathways related to sulforaphane treatment. Molecular targets of sulforaphane could improve the response of actual cancer treatments and triggered autophagy, apoptosis, DNA damage, and/or cell cycle arrest.

**Figure 2 medicina-58-01377-f002:**
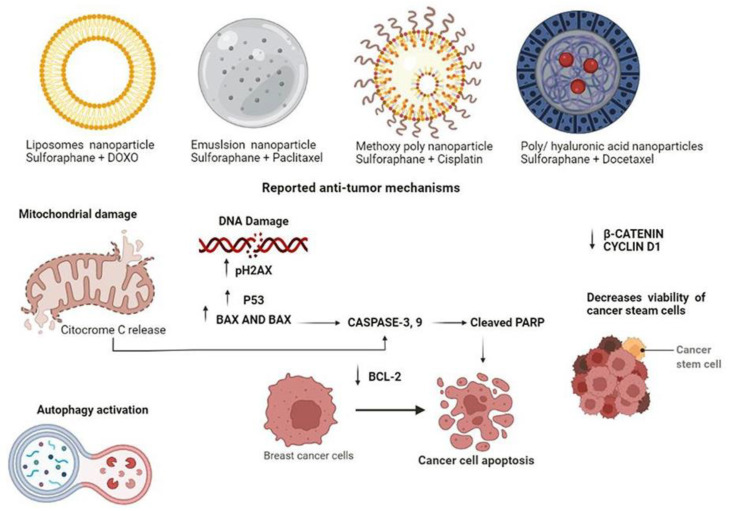
Reported anti-tumour mechanisms of sulforaphane as adjuvant. Mechanisms reported for liposome nanoparticles microemulsion, methoxy poly (ethylene glycol)-poly (lactide-co-glycolide) and poly (D, L-lactideco-glycolide)/hyaluronic acid nanoparticles loaded with sulforaphane and chemotherapeutic agents such as DOX, PTX, cisplatin, and DTX. Among the biological mechanisms of these nanoparticles are the activation of apoptosis through the mitochondrial pathway, autophagy, and the decrease in the viability of cancer steam cells.

**Table 1 medicina-58-01377-t001:** Precursors and products of sulforaphane.

Precursor	Molecule Products	Conditions or Necessary Enzyme	Reference
Glucoraphanin	Thiohydroxamate-O-sulfonate and glucose	Myrosinase	Kensler, et al. [26] Cottaz, et al. [27] de Oliveira [28]
Stable Isothiocyanates			
Thiohydroxamate-O-sulfonate	Sulforaphane	pH 6–7, microbial thioglucosidase	Kensler, et al. [26] Holst [29] Faulkner [30]
	Erucin		
	Iberin		
Unstable isothiocyanates			
β-hydroxylated Isothiocyanate	Goitrin	pH 6–7 and spontaneous cyclisation	Ludikhuyze, et al. [25]
Indolylmethyl-isothiocyanate	3,3-Diindolylmethane	Trimeres, tetramers, etc. of Indole-3-carbinol	Probrazhenskaya [31]
	Ascorbigen	Ascorbic acid presence	Faulkner [30] Preobrazhenskaya [31]
Nitrile aliphatic and aromatic, indole and β-OH-nitriles			
Thiohydroxamate-O-sulfonate	Crambene	pH 3–7 and the presence of nitrile-forming factor	Kong, et al. [32]
Others			
Thiohydroxamate-O-sulfonate	Epithioalkylnitrile	Presence of double bound at Radical, epithiospecifer protein and Fe^2+^	Cottaz, et al. [27] Kong, et al. [32] Foo, et al. [33]
	Thiocyanate	Not clear	Pradhan [34]

**Table 2 medicina-58-01377-t002:** Sulforaphane effect in breast cancer.

Subjects	Sulforaphane Dosage	Anticancer Effect	Genes Targets		Reference
MDA-MB 231.	40 µm	Apoptosis and necrosis	ZEB1, Claudin, and Fibronectin		Mokhtari et al. [40]
BalbC/nude mice	50 mg/kg	Apoptosis and decrease in tumour volume.	ALDH1A1, NANOG, GDF3 and FOXD3		Castro et al. [55]
MC7-cells and MDA-MB-231	5.0 μM	Cell cycle arrest	CCND1 and CDK4		Royston et al. [56]
MCF-7 and MDA-MB-231	4.05 M, 19.35 M, and 16.64 M	Cell cycle arrest	cell cycle arrest		Lewinska et al. [57]
MDA-MB-231, SK-BR-3, and MCF-7	20 ng/mL	Cell cycle, suppression of osteolytic bone resorption	RUNX2 and NF-κB1		Pore et al. [58]
MDA-MB-231, SK-BR-3, and MCF-7	7.5–30 µM	Migration and invasion of breast cancer cells	RAF/MEK/ERK pathway		Zhang et al. [59]
MDA-MB-468	1.8 µM	Cell cycle arrest	EGFR, BCL2, BAX and Akt/mTOR pathway		Yasunaga et al. [60]
Nanoparticles in sulforaphane treatment					
Subjects	Sulforaphane dosage	Anticancer Effect	Genes Targets	Nanoparticles	References
MDA-MB-231 and MCF7	487.5 mg/g + 9.375 mg/g of DOX	Inhibit tumour cell growth, ROS generation, mitochondrial damage and autophagy	Did not report	Liposome nanoparticles	Mielczarek et al. [61]
MDA-MB-231 and MCF-7	8.7 µM and 500 nM of paclitaxel (PTX)	Growth suppression	Did not report	Micro-emulsion nanoparticle	Kamal et al. [62]
MCF-7	48.97 µM and cisplatin at 72.59 µM	Induction of apoptosis through DNA damage.	p-H2AX, p53, PARP and Bcl-2	Methoxy poly (ethylene glycol)-poly nanoparticle	Xu et al. [63]
Breast cancer CD44+ CD24-	350 µg and docetaxel (DTX) 5 nM	Cell growth inhibition	B-catenin and Cyclin D1	Poly (D, L-lactideco-glycolide)/hyaluronic acid nanoparticles	Huang et al. [64]
MCF-7 and BT-474	48 nM and metformin 42 nM	Cell viability decrease and apoptosis	WNT-1, B-catenin, and CD44.	Carbon nanoparticles	Keshandehghan et al. [65]
Adult male Wistar rats	10 mg/kg	Cell viability decrement	Nrf2 pathway	Selenium nanoparticle	Krug et al. [66]
Adult male Wistar rats	10 mg/kg	Induction of ROS production	Did not report	flower-like tellurium nanoparticles	Krug et al. [67]

## Data Availability

Not applicable.

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
