# Peer review of "Molecular Pathways Related to Sulforaphane as Adjuvant Treatment: A Nanomedicine Perspective in Breast Cancer"

_medicina, 2022, doi:10.3390/medicina58101377_

Round 1
Reviewer 1 Report
Sulforaphane is one of the most popular and one of the most tested natural isothiocyanate, which can be found in cruciferous vegetables like broccoli, radish or Brussels sprouts. Sulforaphane shows many biological activity, which one of them is anticancer activity. Sulforaphane is able to simultaneously modify many cellular targets associated with cancer development, including DNA protection, by inhibiting the activity of mutagenic factors (phase I) and activation of phase II factors responsible for detoxification, inhibiting the proliferation of cancer cells and activating apoptosis, thereby limiting the process of multiplication of mutated cancer cells, and inhibiting the process of neogenesis and metastasis.
In this review article authors discuss the properties and benefits of sulforaphane in cancer therapy, as well as their ability to form complexes with nanomolecules and chemotherapeutic agents that synergize the antitumor response in breast cancer cells.
In my opinion this review is written succinctly, correctly, and is pleasant to read. I have no major objections to the review, however I have some minor comments:
- Line 44: availability[5–7]stable – the bracket with links at the end of line.
- Line 58: it should be lavto-ovo-vegetarian
- Line 60: it should be mahanine
- Line 87: I think it should be isothiocyanates not Sulforaphane. Sulforaphane is obtained in reaction of glucoraphanin with myrosinase. In this paragraph (Line 81-87) you write about many different glucosinolates, so you can’t write at the end that these molecules generating Sulforaphane. They generating isothiocyanates. You also forget about Lossen rearrangement in synthesis of isothiocyanates. Please complete this.
- Line 89: this glucosinolate is called glucoraphanin
- Table1. I think it should be extra line between Precursor and Glucoraphanin with Sulforaphane.
- Line 133 and 134: the sentence the activation of p34cdc2 kinase is two times. Should it be like that?
- Line 149: balance (space) [41].
- Line 151: sulforafane small first letter
- Line 168: concentration is M not m. It shouldn’t be 50 μM?
- Line 175: 50 μM
- Line 197: 20 μM/L
- Line 221: Figure 1
- Line 228: pathways
- Line 236: sulforaphane and 40 μM
- Line 239: 40 μM
- Line 241: 15 μM
- Line 256: MDA-MB-231 (space) cell line
- Line 315: Fe3O4, PEGylated Fe3O4
- Line 316: Fe2+ and Fe3+
- Line 343: Figure 2
- Line 357: 5 nM
- In references 1, 4, 8, 9, 15, 16, 19, 24, 40, 41, 42, 44, 46, 48, 52, 55, 57, 59, 60, 62, 66, 67, 69, 70 and 71 replace full name of journals with abbreviations.
Overall, I recommend review article in Medicina if the above comments will be included in the manuscript.
Author Response
Dear Reviewer:
Good afternoon. We thank you for your comments, which were dealt with in the best possible way (attached document).
Regards!

Reviewer 2 Report
This work deals with overviewing the investigation of molecular pathway of sulforaphane such as PI3K/AKT/mTOR associated with breast cancer, to better understand signalling mechanism of it.
The sulforaphane structure and features are studied regarding breast cancer therapy process.
REMARKS
1 Follow the upgrade of the last paragraph in Introduction including citing of the current work dealing with chemotherapy drug agent study.
“In general, certain bioactive and other compounds can suppress or promote the expression of certain key genes in breast cancer by modulating their transcription, blocking or activating signalling pathways involved in proliferation, differentiation, and apoptosis, and counteracting the toxic effects of chemotherapy drugs [1,9–12, https://doi.org/10.1515/biol-2019-0070].
2 Provide an information table that would include the most recent works dealing with sulforaphane and revealing other molecular pathways associated with cancer. This table would also include information of the main outcomes of these works.
3 In conclusion, provide a short information of what would the authors do in the future as for the breast cancer investigation regarding molecular pathways and therapies associated with bioactive compounds.
Author Response
Dear reviewer:
We thank you for your comments on the review, which were addressed in the best possible way (attached document).
Best regards!

Round 2
Reviewer 2 Report
The suggested paper deals with study of chemotherapy drugs in breast cancer, so it fits to the topic of using chemotherapy in general. I suggest to follow my remark literally.
Round 3
Reviewer 2 Report
I agree to publish the paper at the current form